

# Endoplasmic reticulum stress and the protein degradation system in ophthalmic diseases

Jing-Yao Song[1], Xue-Guang Wang[2], Zi-Yuan Zhang[1], Lin Che[1], Bin Fan[1] and Guang-Yu Li[1]

[1] Department of Ophthalmology, Second Hospital of Jilin University, ChangChun, China
[2] Department of Traumatic Orthopedics, Third People's Hospital of Jinan, Jinan, China

## ABSTRACT

**Objective**. Endoplasmic reticulum (ER) stress is involved in the pathogenesis of various ophthalmic diseases, and ER stress-mediated degradation systems play an important role in maintaining ER homeostasis during ER stress. The purpose of this review is to explore the potential relationship between them and to find their equilibrium sites.
**Design**. This review illustrates the important role of reasonable regulation of the protein degradation system in ER stress-mediated ophthalmic diseases. There were 128 articles chosen for review in this study, and the keywords used for article research are ER stress, autophagy, UPS, ophthalmic disease, and ocular.
**Data sources**. The data are from Web of Science, PubMed, with no language restrictions from inception until 2019 Jul.
**Results**. The ubiquitin proteasome system (UPS) and autophagy are important degradation systems in ER stress. They can restore ER homeostasis, but if ER stress cannot be relieved in time, cell death may occur. However, they are not independent of each other, and the relationship between them is complementary. Therefore, we propose that ER stability can be achieved by adjusting the balance between them.
**Conclusion**. The degradation system of ER stress, UPS and autophagy are interrelated. Because an imbalance between the UPS and autophagy can cause cell death, regulating that balance may suppress ER stress and protect cells against pathological stress damage.

# INTRODUCTION

The endoplasmic reticulum (ER) is a highly dynamic and important organelle of eukaryotic cells that has many functions, such as mediating free calcium storage, regulating lipid/sterol synthesis, and participating in the synthesis, processing, and transportation of proteins (*Fregno & Molinari, 2018*). When the internal environment of the ER is destroyed, the accumulation of improperly folded proteins therein eventually leads to ER stress (*Li et al., 2017a*). In order to inhibit ER stress and coordinate the recovery of ER function, cells have integrated signaling systems, including unfolded protein response (UPR) and the ER-associated degradation (ERAD) pathway (*Fujita et al., 2007*).

Corresponding author
Guang-Yu Li, liguangyu@aliyun.com

During ER stress, the UPR is activated and performs physiological functions that include enhancement of protein folding ability, stasis of translation of most proteins, and acceleration of protein degradation (*Kroeger et al., 2019*; *Labbadia & Morimoto, 2015*). Moreover, ERAD pathways composed by ubiquitin (Ub)—proteasome-dependent and autophagy—lysosome-dependent ERAD are also activated to participate in the removal of improperly folded proteins in order to restore the function of the ER (*Schroder & Kaufman, 2005a*).

Although ER stress, autophagy, and the ubiquitin proteasome system (UPS) have been fully studied in ophthalmic diseases, the relationship among them requires further examination. The purpose of this article is to explore the relationship between them in order to find new opportunities for future research directions and treatment of diseases. We suggest that diseases caused by ER stress can be blocked by regulating the balance between autophagy and the UPS, and especially by removing the pathogenic factor that results in ER stress that cannot be effectively removed.

## Survey methodology

This review focuses on the relationship between ER stress, autophagy, and the UPS and their interactions in ocular diseases. Academic articles were searched in journal databases such as Web of Science, PubMed, and ER stress, autophagy, UPS, ophthalmic disease, and ocular were the search terms used for article research. The inclusion criteria for the selected articles required that articles be related to ophthalmic diseases and focused on the relationship between ER stress, autophagy, or the UPS.

## ER stress and the UPR

The ER is an important site for protein synthesis, and therefore, if the function of the ER is disrupted by various pathological factors, the excessive accumulation of unfolded or misfolded proteins in the ER may eventually lead to ER stress. Many pathological conditions can lead to ER stress, such as hypoxia, oxidative stress, aging, or metabolic disorders (*Lenox et al., 2015*; *Rozpedek et al., 2016*; *Rutkowski & Hegde, 2010*; *Zhu et al., 2018*). When ER stress happens, the UPR is activated as a protective mechanism to restore the balance of the ER environment. The UPR involves 3 ER transmembrane proteins: activated transcription factor 6 (ATF6), ER to nuclear signaling 1 (ERN1; also known as inositol-requiring enzyme 1 [IRE1]), and eukaryotic translation initiation factor 2–kinase 3 (EIF2AK3; also known as protein kinase R–like endoplasmic-reticulum kinase [PERK]) (*Ron & Walter, 2007*). Under normal physiological conditions, these proteins bind to the 78 kDa glucose regulatory protein (Grp78; also known as binding immunoglobulin protein [BiP]) in the ER with lumen. During ER stress, BiP is separated from the sensor so that the UPR signals become activated. IRE1 then becomes phosphorylated, which activates endoribonuclease activity, splices the 26-nucleotide (nt) sequence from X-box binding protein (XBP1) messenger ribonucleic acid (mRNA) and produces functional XBP1(S), which is transferred to the nucleus and activates transcription of the genes encoding the ER chaperone and ERAD (*Wakabayashi & Yoshida, 2013*). PERK is phosphorylated and activated, which in turn phosphorylates eukaryotic translation initiation factor 2 (eIF2),
thereby inhibiting protein translation and reducing protein synthesis (*Dan et al., 2017*). ATF6 is transported to the Golgi in the form of vesicles and is then cleaved by protease (site 1 and site 2 protease [S1P and S2P]) to produce a transcriptionally active polypeptide (*Chen, Shen & Prywes, 2002*). Activated ATF6 translocates to the nucleus and activates the gene transcription of proteins such as ER chaperones that increase ER protein folding (*Yamamoto et al., 2007*).

Interestingly, recent studies have shown that the three signaling pathways of the ER transmembrane proteins may interfere with each other. For example, the IRE1α and PERK pathways are not mutually independent, because knocking out IRE1α can alter the PERK pathway and also lead to decreased eIF2α expression (*Storniolo et al., 2018*). ATF6 is associated with IRE1, and ATF6 knockdown can result in unchecked IRE1 reporter activity that increases the splicing of XBP1 (*Franziska et al., 2018*).

## ER stress and the UPS

The UPS is an important protein degradation pathway in eukaryotic cells. Protease and ubiquitin (Ub) in collaboration are responsible for non-lysosomal protein hydrolysis, which may remove abnormal proteins and prevent the accumulation of nonfunctional and harmful proteins in cells, therefore maintaining cellular homeostasis (*Angele et al., 1999*; *Coux, Tanaka & Goldberg, 1996*). The UPS may participate in a variety of biological processes, such as cell cycle, transcription, signaling, trafficking, and protein quality control (*Jiang, Zhao & Qiu, 2018*; *Rousseau & Bertolotti, 2018*). Ub, the smallest protein found in all eukaryotic cells (8 kDa), covalently conjugates many proteins to label them for downstream effector recognition (*Cohen-Kaplan, Ciechanover & Livneh, 2017*). Ub plays an important role in cells, including DNA repair, kinase activation, secretion, and protein transport in endocytic pathways (*Kostova, Tsai & Weissman, 2007*).

The conjugation of Ub protein with a substrate is a multi-step reaction that requires the participation of many enzymes and consumes energy (*Glickman & Ciechanover, 2002*; *Pickart & Eddins, 2004*). Targeted proteins undergo Ub–proteasome degradation by the 26S proteasome, which is highly conserved as a 2.5-MDa complex responsible for selective ATP-dependent degradation of ubiquitinated proteins in eukaryotic cells (*Zwickl, Voges & Baumeister, 1999*). It is composed of 2 large subcomplexes consisting of a 28-subunit 20S protease and a 19-subunit PA700 compound (also called a 19S complex or an S-regulating particle) (*Liu & Jacobson, 2013*). The 20S proteasome is responsible for substrate degradation, and its 19S subunit assists in degradation primarily by recognizing the substrate (*Glickman, 2000*). The 19S complex plays an important role in processing ubiquitinated substrates because it binds, ubiquitinates, and unfolds ubiquitinated protein, which is then transferred to the proteolysis chamber of the 20S proteasome for degradation (*Zwickl, Voges & Baumeister, 1999*).

As an important branch of the ERAD pathway, most of the unfolded and misfolded proteins in the ER are degraded by the UPS pathway (*Nakatsukasa & Brodsky, 2008*; *Zattas & Hochstrasser, 2015*). The UPS plays an important role in maintaining ER homeostasis, and impaired Ub–proteasome function can lead to ER stress (*Shruthi et al., 2016*). Because the UPS can remove proteins from the ER, it is vital to be able to correctly identify improperly

folded proteins in the ER. There are at least two monitoring mechanisms for ER protein folding, consisting of one for the luminal domain (soluble or membrane proteins), and the other for the cytoplasmic domain (membrane proteins) (*Vashist & Ng, 2004*). The UPS removes proteins in the ER through four tightly coupled steps: (1) substrate selection, (2) retro translocation to the cytosol, (3) C-conjugated covalent polyubiquitination, and (4) proteasome degradation (*Olzmann, Kopito & Christianson, 2013*).

Substrate degradation begins with molecular chaperones, which identify proteins to be degraded by detecting abnormal disulfide bonds or hydrophobic fragments exposed by unassembled protein complexes (*Vembar & Brodsky, 2008*). In addition, another marker used to identify misfolded proteins is the presence of high mannose (Man5–8GlcNAc2) glycan (*Mallinger et al., 2012*). The substrates are targeted to the retrotranslocation machinery and then translocated to the cytoplasm through the retrotranslocation channel (*Vembar & Brodsky, 2008*).

In addition, ubiquitination is a complex process that requires the participation of three enzymes: ubiquitin activating enzyme E1 (ubiquitin activating enzyme), ubiquitin binding enzyme E2 (ubiquitin-conjugating enzyme, E2), and E3 (ubiquitin ligase) (*Pickart, 2001*). At first, E1 forms a high-energy thioester bond with ubiquitin in an ATP-dependent manner, activating the ubiquitin molecule. Then, the activated ubiquitin molecule is transported to E2, acquires the function of recognizing the target protein, and finally binds the target protein under the catalysis of E3 (*Glickman & Ciechanover, 2002*). After repeated enzymatic reactions, the polyubiquitin chain binds to the target protein, which is recognized by the 26S proteasome, and then degraded (*Hershko & Ciechanover, 1998*; *Voges, Zwickl & Baumeister, 1999*).

There are many types of E2 enzymes involved in ubiquitination, but not all E2 types are involved in ERAD. The three types involved in ERAD are ubiquitin-conjugating enzymes J1, J2, and G2 (UBE2J1, UBE2J2, and UBE2G2, respectively) (*Christianson & Ye, 2014*). Similarly, not all E3 enzymes are involved in ERAD. In yeast, Doa10p and Hrd1p are E3 ligases, and they have participated in the degradation of all substrates that have been studied (*Cui et al., 2012*). However, E3 ligases found in mammals, such as Hrd1/synoviolin, gp78, TEB4/MARCHVI, RNF5, HRD1, RNF-12, and RNF185, are involved in protein degradation in the ERAD pathway (*Darom, Bening-Abu-Shach & Broday, 2010*; *El Khouri et al., 2013*; *You et al., 2016*). It was reported that the UPS is regulated by the UPR pathway; for example, the PERK signal can increase the expression of RNF-121 to further enhance the UPS during ER stress (*Darom, Bening-Abu-Shach & Broday, 2010*). In addition, XBP1, the downstream factor of IRE1, is required for Nrf2 expression which is the central regulator of cell-protective genes ubiquitous expressed in cells, while Nrf2 interacts with the Cullin3-based E3 ubiquitin ligase adaptor to promote the proteasome (*Chen et al., 2018*; *Ding et al., 2017*; *Tonelli, Chio & Tuveson, 2018*). This indicates that the Nrf2 factor may be a link by which the IRE1 pathway regulates the UPS. Moreover, rapamycin is the core of proteasome assembly regulation, while ATF6 is essential for mediating ER stress to activate the mammalian target of the rapamycin (mTOR) pathway (*Dylan & Jin, 2018*). Therefore, mTOR may be the intermediate factor for ATF6 to activate UPS.

## ER stress and autophagy

Autophagy is another metabolic pathway that regulates the degradation of long-lived proteins, organelles, and other cellular contents (*Liu et al., 2015*). Autophagy can be divided into macrophagy, microautophagy, and molecular chaperone-mediated autophagy (CMA) (*Bejarano & Cuervo, 2010*; *Mijaljica, Prescott & Devenish, 2011*; *Parzych & Klionsky, 2014*). Macroautophagy is divided into non-selective and selective autophagy. Non-selective autophagy is usually induced by nutrient deprivation and often involve the mTORC1 and protein kinase AMP–activated catalytic subunit alpha (PRKAA)/adenosine monophosphate–activated protein kinase (AMPK) pathways (*Ganley et al., 2009*). Selective autophagy targets specific substrates, including protein aggregates and damaged organelles such as mitochondria and peroxisomes (*Lamark & Johansen, 2012*). Macroautophagy is a continuous process involving the formation of autophagosomes, the fusion of autophagosomes with lysosomes, and the dynamic process of lysosomal degradation (*Baehrecke, 2005*). It is an important metabolic pathway in eukaryotic cells that is often used to resist stress and maintain intracellular homeostasis (*Boya et al., 2016*; *Lin & Kuang, 2014*; *Shi et al., 2013*). In the process of microautophagy, the lysosomal membrane acts as a concave protuberance or membrane, allowing a small portion of the cytoplasmic volume to enter the lysosomal cavity, which degrades the substrate (*Li, Li & Bao, 2012*). Molecular CMA does not require vacuolar formation and is tightly regulated by chaperone heat shock cognate 71 kDa protein (Hsc70) and its receptor, and it is associated with the PERK pathway in ER stress (*Li et al., 2017b*).

Hence, improperly folded proteins are not only degraded through the UPS pathway, but autophagy is also involved in protein degradation, and an increasing number of studies have shown that ER stress may trigger autophagy (*Bachar-Wikstrom et al., 2013*; *Chandrika et al., 2015*). Misfolded proteins and protein aggregates are cleared under stress by autophagy, especially when the other cellular repair and cellular clearance processes, namely molecular CMA and the UPS, fail (*Libby & Gould, 2010*; *Pandey et al., 2007*). In general, ER stress induces autophagy through the IRE1 and PERK signaling pathway (*Ogata et al., 2007*; *Wafa et al., 2013*). The downstream factor JNK is activated through the IER1 signaling pathway and further promotes autophagy during ER stress (*Corazzari et al., 2017*). Phosphorylated IRE1 also activates the MAPK8/JNK1/MAPK9/MAPK10 pathway, thereby upregulating autophagy (*Yan et al., 2018*). In addition, spliced XBP1 is reported to be involved in the activation of autophagy by upregulating the transcription of BECN1 (*Christen & Fent, 2012*). The PERK signaling pathway is activated during ER stress, and its downstream factor eIF2a phosphorylates, while phosphorylated eIF2a activates deoxyribonucleic acid (DNA) damage—inducible transcript 3 (DDIT3)/ATF4, thereby promoting tribbles pseudokinase 3 (*TRIB3*) to induce autophagy by inhibiting Akt1/mTORC1 (*Salazar et al., 2009*; *Tang et al., 2015*). Additionally, ATF4, which is the downstream factor of PERK, may function as a transcription factor regulating the expression of various autophagy-related genes (*Wafa et al., 2013*) (Fig. 1). Studies have shown that ER stress-related autophagy is mainly mediated by the IRE1a and PERK pathways, while the ATF6 signaling pathway can indirectly regulate autophagy by upregulating the expression of XBP1 and CHOP.
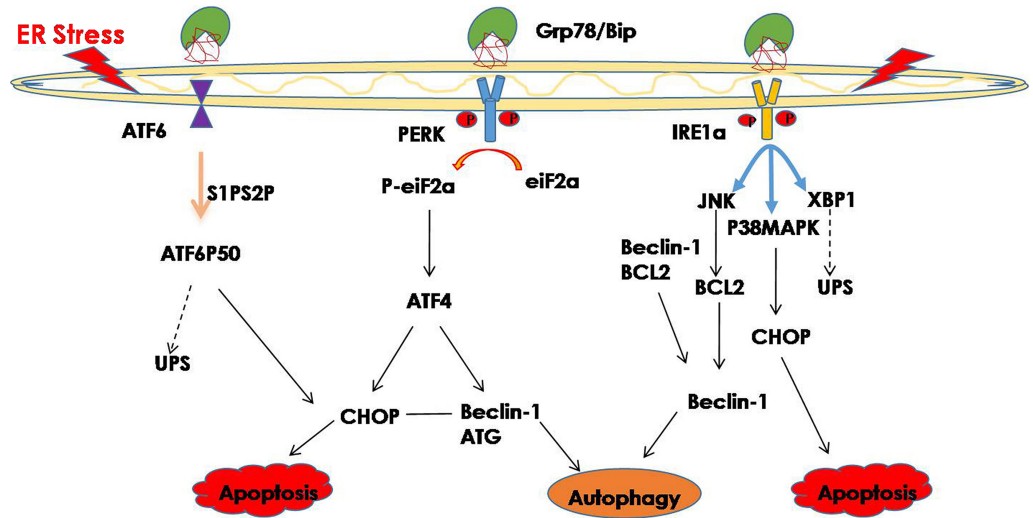

**Figure 1** **ER stress and its degradation pathways.** When ER stress occurs, in order to restore the function of the ER, the UPR is activated, and the UPS and autophagy are activated to suppress ER stress. However, if they still cannot restore the function of the ER, cell death may result.

## UPR and ophthalmic diseases

The UPR is known to be an adaptive cellular response to ER dysfunction that suppresses ER stress and promotes cell survival. Many studies have found that proper UPR response plays an important role in maintaining the normal physiological function of cells (*Sano & Reed, 2013*; *Schroder & Kaufman, 2005b*). Insufficient activation of the UPR response to ER stress is the pathogenic factor of age-related retinal neurodegeneration. Thus, the lack of X-box binding protein 1 (XBP1), which is an important component of the UPR, may accelerate age-related retinal neurodegenerative diseases (*Mclaughlin et al., 2018*). It was also reported that in retinal pigmented epithelium (RPE)-specific XBP1 KO mice showed a 33% reduction in retinal cone cells and reduced the thickness of the outer nuclear layer (ONL), suggesting that XBP1 plays an important role in maintaining ER homeostasis and normal RPE cell function (*Zhong et al., 2012*). Moreover, it was shown that PERK activation is a protective response that increases the survival of photoreceptors in a P23H-1 transgenic rat model, indicating that UPR plays an important role as the first line of defense against protein-toxic cellular stress (*Athanasiou et al., 2017*). In addition, the ATF6 pathway is crucial to human color vision, and ATF6 mutation causes autosomal-recessive color blindness (*Ansar et al., 2015*).

UPR plays an important role in maintaining the normal physiological functions of cells under ER stress; however, if UPR still cannot suppress ER stress, it may result in cell death. Thus, if ER stress cannot be effectively controlled, phosphorylated IRE1α forms a complex with TRAF2 and ASK1 and activates the downstream factor JNK, which may activate caspase-12 to promote apoptosis (*Ron & Hubbard, 2008*). PERK activation further increases the expression of ATF4 and CHOP, promotes transcription of genes involved in oxidative stress and apoptosis, and further leads to cell death (*Lu, 2004*;

*Rzymski et al., 2009*). The activated ATF6 translocates into the nucleus, where it binds the ER stress response elements to activate target genes, including XBP-1 and CHOP, which directly or indirectly result in cell death (*Adachi et al., 2008*; *Guo et al., 2014*; *Hirsch et al., 2014*).

An increasing number of studies have shown that ER stress is a factor in the pathology of many ophthalmic diseases, such as chronic glaucoma, glucocorticosteroid-induced glaucoma, cataract, DR, optic-nerve (ON) degeneration, and AMD (*Doh et al., 2010*; *Elmasry et al., 2018*; *Ojino et al., 2015*; *Palsamy & Shinohara, 2017*; *Salminen et al., 2010*; *Zhou, Bennett & Shiels, 2016*; *Zode et al., 2014*). For this reason, suppressing ER stress and quickly restoring the ER homeostasis play vital roles in the treatment of those diseases. For example, a recent study indicated that ER stress is detrimental for retinas in the early stages of DR, and suppressing ER stress may protect retinas against visual deficits caused by hyperglycemia (*Raji et al., 2018*). Moreover, in a mice model of TON induced by optic nerve crush (ONC), ER stress results in RGC death, however suppressing ER stress through GRP78 overexpression is an effective way to protect RGC from death (*Ha et al., 2018*). This suggests that reducing the pathological factors that result in ER stress and maintaining cell homeostasis are key factors in sustaining cell physiological functions. Therefore, UPS and autophagy, as important mechanisms of ERAD, play vital roles in restoring ER homeostasis during ER stress.

## UPS and ophthalmic diseases

The UPS is essential for eye health and is involved in organ development and maintenance of lens function (*Liu, 2015*; *Wride, 2011*). In addition, it was reported that the proteasome can accelerate protein renewal and efficiently accelerate the degradation of rhodopsin T17M mutant (*Jiang, Xiong & Xia, 2014*). Inadequate UPS function is involved in the development of ophthalmic diseases. In glaucoma, decreased ubiquitination in the optic nerve may increase the level of proapoptotic proteins that are normally degraded by proteasomes, leading to axonal degeneration after increased intraocular pressure (IOP) (*Dibas et al., 2008*). However, overactivation of the UPS also impairs eye health. For example, tumor necrosis factor (TNF) destroys the interstitial connections in human corneal fibroblasts in a manner dependent on the UPS degradation of connexin 43 (Cx43) (*Kimura & Nishida, 2010*). A study reported that UPS participated in the degradation of rhodopsin and impairs visual function during retinal inflammation (*Ozawa et al., 2008*). Therefore, it was reported that UPS is involved in LPS-induced rat endotoxic uveitis (EIU); intravitreal resolvin D1 (RvD1) can inhibit uveitis by reducing the local level of Ub–proteasome (*Rossi et al., 2015*).

Given that the UPS is an important part of the ERAD pathway, and that it is involved in removing intracellular proteins, the UPS plays an important role in the inhibition of ER stress-mediated ophthalmic diseases. The importance of the UPS has been demonstrated in a large number of diseases associated with protein misfolding (*Guerriero & Brodsky, 2012*). It was shown that ER stress is the pathogenic factor involved in granular corneal dystrophy type 2 (GCD2), while the UPS activated by melatonin can accelerate the degradation of TGF-$\beta$-inducible protein (TGFBI) to suppress ER stress, and thus prevent the death of

GCD2 cells (*Choi et al., 2017*). In addition, a growing number of studies have shown that if the UPS cannot restore the homeostasis of ER during ER stress, many ophthalmic diseases may occur. For example, in pseudo-exfoliation (PEX) disease, ER stress is overactivated, and the UPS is unable to remove harmful substances, resulting in secondary glaucoma (*Hayat et al., 2019*). Analysis of retinal proteins in patients with high IOP showed the existence of ER stress, and if the UPS cannot inhibit overactivated ER stress, this could eventually lead to retinal damage (*Yang et al., 2015*).

## Autophagy and ophthalmic diseases

Under normal physiological conditions, autophagy, which plays an important role in maintaining normal cell function, is maintained at a relatively low level. Neuronal cells, for example, control cytoskeleton and organelle turnover through the autophagy process, allowing neurons to survive and regenerate after distal axon dissection or nerve suture (*David, Crish & Inman, 2015*; *Tatiana et al., 2018*). Some studies have shown that autophagy deficiency is a pathological factor leading to many ophthalmic diseases such as corneal opacification, elevated IOP, retinal dystrophy, mucopolysaccharide storage disease type VI, and AMD (*Claudepierre et al., 2010*; *Golestaneh et al., 2017*; *Karnati et al., 2016*; *Lőrincz et al., 2016*). Moreover, it was reported that exposure of *ex-vivo* mice retinal explants to high glucose resulted in the death of retinal neuronal cells, while treatment the explants with octreotide may protect neuronal cells against high glucose damage by enhancing autophagy (*Amato et al., 2018*). Although proper autophagy is beneficial to cell survival under stress, overactivated autophagy may lead to cell death, which is called autophagic cell death (ACD) (*Liu & Levine, 2015*; *Vegliante & Ciriolo, 2018*). A study demonstrated that the over-activated autophagy lead to the death of photoreceptors and inhibition of autophagy with 3MA may protect photoreceptors against photodamage (*Zhang et al., 2014*). Thus, as a double-edged sword, autophagy may either promote cell survival or lead to cell death, depending on the duration and intensity of pathology.

In general, autophagy, as another component mechanism of the ERAD pathway, is a survival mechanism to protect cells against stress, and a large number of studies have shown that autophagy can suppress ER stress and attenuate the pathological damage caused by stress. In glaucoma, enhanced ER stress-mediated autophagy may accelerate myosin clearance in trabecular meshwork cells, thus protecting them against damage. Sulforaphane (SFN) reduces the incidence of posterior cataracts by increasing ER stress-mediated autophagy (*Liu et al., 2017*). It was also reported that neurons in the lesioned cortex undergo apoptosis after traumatic brain injury, however, treatment with sevoflurane may enhance ER stress-mediated autophagy and inhibit neuronal apoptosis (*He et al., 2018*). However, ER stress-mediated autophagy also acts as a double-edged sword. For example, It has been shown that in diabetic retinopathy ER stress-mediated autophagy caused by a low concentration of oxidized glycosylated low-density lipoprotein (HOG-LDL) may attenuate the loss of peripheral blood cells, while prolonged ER stress-mediated autophagy caused by a higher concentration of HOG-LDL may promote the death of peripheral blood cells (*Fu et al., 2016*). Hence, excessive ER stress-induced autophagy may also lead to cell death. It was shown that the protective effect of mini- αA on NaIO3-induced retinal

degeneration was achieved by inhibiting ER stress and autophagy (*Zhang et al., 2015a*). A recent study showed that in a mouse model of retinal degeneration induced by a P23H rhodopsin gene mutation, the accumulation of misfolded proteins in retinal photoreceptor cells activated ER stress and excessive autophagy, while inhibition of autophagy via deleting the autophagy-activating gene Atg5 decreased photoreceptor death and improved retinal function (*Yao et al., 2018*). Therefore, whether ER stress-induced autophagy is protective or damaging depends on disease conditions.

## The important role of balance between autophagy and UPS during ER stress

Both the UPS and autophagy play important roles in maintaining the balance of cellular proteins, and each has its own advantages. The UPS is responsible for the degradation of both short-lived proteins and misfolded proteins, while autophagy can degrade misfolded proteins and damaged organelles (*Li et al., 2016*). It was found that there is a certain relationship between the UPS and autophagy. It is known that sequestosome 1 (*SQSTM1*) is a multitasking bridging protein that regulates multiple signaling pathways, and the UPS and autophagy are correlated with each other through P62 protein (*Jorge & Diaz-Meco, 2009*; *Milan et al., 2015*). In addition to p62, other adaptors, such as neighbors of type 1 breast cancer (NBR1), can also recognize ubiquitinated substrates and localize them to autophagosomes (*Cohen-Kaplan et al., 2016*). In general, Ub ligase E3 is mainly degraded and regulated by proteasomes or by the recycling of its own ubiquitination. However, a recent study demonstrated that etoposide-induced protein 2.4 homolog (EI24) recognizes the RING domain existing in most E3 ligases and degrades them via the autophagic pathway (*De Bie & Ciechanover, 2011*; *Nam et al., 2017*). In addition, autophagic inhibition impairs the UPS function and leads to ER stress (*Zhang et al., 2015b*). It was shown that the functions of autophagy and the UPS are complementary in some conditions, and passive regulation of the functions between them is necessary to maintain cell protein homeostasis (*Jung et al., 2019*) (Fig. 2).

A growing number of studies have shown that the UPS and autophagy may restore cellular homeostasis through mutual regulation. For instance, low levels of proteasome inhibitors in the treatment of oxidative stress injury of RPE cells can inhibit the PI3K/AKT/mTOR pathway and activate autophagy, thus protecting RPE cells against oxidative damage (*Tang et al., 2014*). In addition, it was reported that inhibition of autophagy, especially in the case of adequate nutrition, can enhance the activity of proteasomes, which are activated as a compensatory form of protein degradation (*Wang et al., 2013*). Moreover, Zacks et al. reported that in a mouse model of retinal degeneration caused by a gene mutation in P23H rhodopsin, ER stress-related autophagy led to photoreceptor death, while the treatment of P23H mice with selective phosphodiesterase-4 inhibitor (rolipram) to increase proteasome activity could effectively inhibit ER stress-related autophagy and reduce the rate of retinal degeneration (*Qiu et al., 2019*). Therefore, the balance between UPS and autophagy is very important, and these two systems have irreplaceable effects on cellular health.

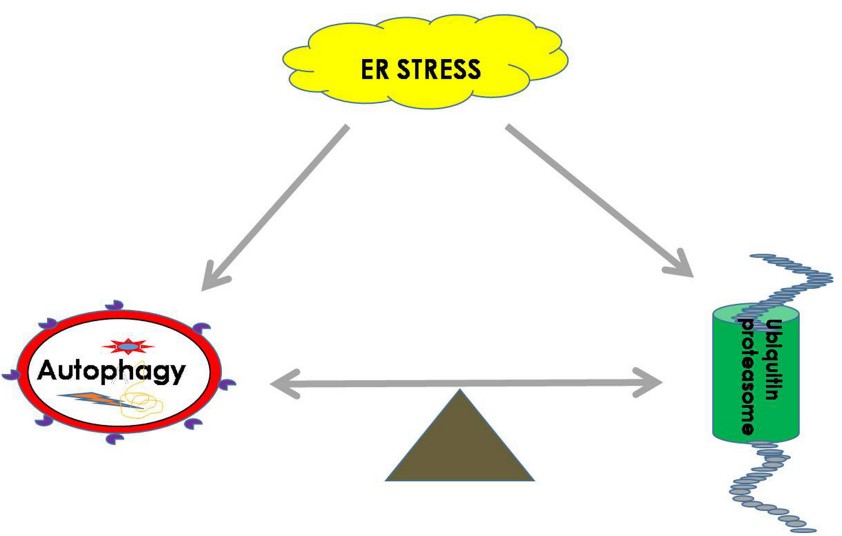

**Figure 2  ER homeostasis can be achieved by balancing the UPS and autophagy pathways during ER stress.** During ER stress, the UPS and autophagy will be activated to remove harmful substrates such as misfolded proteins or protein aggregates to maintain the normal function of the ER. The balance between the UPS and autophagy is extremely important for restoring cell homeostasis.

## CONCLUSION

Autophagy and the UPS are normal phenomena that manage the health of the living eukaryotic cell. However, deficiency and excessive activation of both autophagy and proteasomes are not conducive to cellular health. Many diseases are related to ER stress, and the UPS and autophagy play an important role in suppressing ER stress and maintaining ER homeostasis. We should take this information into consideration while also removing pathogenic factors, especially pathogenic factors such as genetic diseases that cannot be removed by current medical treatments. Then, we may be able to inhibit diseases by simultaneously regulating autophagy and the UPS in order to achieve intracellular homeostasis.

## ACKNOWLEDGEMENTS

We thank LetPub for its linguistic assistance during the preparation of this manuscript.

### Funding

The study was funded by the National Natural Science Foundation of China (No. 81570864) and the Natural Science Foundation of Jilin Province (No. 20160101004JC; No. 20160414045GH; No. 2016J041). The funders had no role in study design, data collection and analysis, decision to publish, or preparation of the manuscript.

## Grant Disclosures

The following grant information was disclosed by the authors:
National Natural Science Foundation of China: 81570864.
Natural Science Foundation RL of Jilin Province: 20160101004JC, 20160414045GH, 2016J041.

## Competing Interests

The authors declare there are no competing interests.

## Author Contributions

- Jing-Yao Song analyzed the data, prepared figures, authored drafts of the paper, approved the final draft.
- Xue-Guang Wang and Lin Che collected the data, drafted the paper, approved the final draft.
- Zi-Yuan Zhang and Bin Fan collected the data, prepared figures, approved the final draft.
- Guang-Yu Li analyzed the data, reviewed drafts of the paper, approved the final draft.

## Data Availability

This is a literature review article and did not generate raw data.

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
