# Peer review of "Endoplasmic reticulum stress and the protein degradation system in ophthalmic diseases"

_PeerJ, doi:10.7717/peerj.8638_

## Round 0.1 · original submission · Major Revisions

I would urge you to address the relatively minor issues raised by the reviewers 1 and 2; but I do tend to agree with reviewer 3 that some attention to the overall arrangement of this review is needed. I would urge you to consider carefully the logic and flow in parts: there are some nice ideas and concepts here but at times they are tricky to identify. I think the final paragraph of reviewer 3's comments is particularly germane.

As this represents a substantial editing, I have selected a major revision, but I do not regard this as a huge amount of additional content being required - rather a focus and sharpening. I do hope you will be able to do this and send us a revised version.

·

Basic reporting

Song et al. reviewed the literature about ER stress and protein degradation system associated ocular diseases. It is a quite complicated topic, the authors managed to put related information together and draw a clear conclusion.

Experimental design

Well designed.

Validity of the findings

Cite related literature and structure related information together.

Additional comments

In general, the authors tried to collect all related background information and organise these information and present the data in a clear way. However the whole structure needs some changes, though I understand this is a quite broad and complicated topic. I would suggest the authors present the all related background information first, for example the sections of 'Understanding the function of the degradation system during ERS' and 'ERS and ub-proetasome response' need to be ahead of other section which describe the involvement of ocular disease. There are a lot of small errors, particularly the citation of references in the main text.

·

Basic reporting

In this review, Song et al. describe the relationship and cross talk between two important cellular events, ER stress and protein degradation. They describe in detail how protein misfolding in the ER triggers a series of adaptive signalling cascades which extends beyond the ER and is aimed at restoration of cellular proteostasis and homeostasis.
Whilst some recent reviews have highlighted the roles of the UPR effectors and the protein degradation pathways in eye diseases (see for example Kroeger H et al., FEBS J, 2019; Chan P et al., Brain Res, 2016; Chai P et al., Int J Biol Sci 2016; Zhu and Du, Int J Ophthalmol, 2018), Song et al., draw attention to the specific roles and interrelationship between ER stress, the UPR and the cellular protein degradation machineries in the pathology of several ocular diseases.
The review therefore addresses an important and rapidly evolving area of research and would be of broad interest.

Experimental design

Song et al., appear to have extensively researched the subject matter, and show a good grasp of existing literature, past and present.
The sources are properly cited, save for what could be an editing error; some references are just inserted in the text and some in superscript. That said, the authors rely too heavily on citing review articles.
In terms of organisation, large parts of the review appear to be disjointed; some subsections and paragraphs are introduced without context. Overall, the structural organisation could be clearer and logical.

Validity of the findings

Song et al., have presented a well-researched review of the literature in a rapidly developing area of research. However, the authors could do more to highlight areas and direction of future research and perhaps suggest how our understanding of the interplay between ERS and protein degradation pathways could impact our understanding of ophthalmology.

Additional comments

Abstract
The first sentence should be rephrased, ‘cause of the occurrence’ in line 32

Introduction
In general, the introduction is well written and succinctly describes the subject matter. It also clearly addresses the set objectives and why the review would be of general interest.

The authors should seek a better word/phrase to replace ‘broken’ in line 54.
The authors should replace ‘chamber’ in line 63, with lumen
‘Two pathways associated with…’ line 66-67, should be a new paragraph
Line 95…’that are generally expressed in cells’ do the authors mean ubiquitous expression? This sentence should be rephrased
Line 98…’Studies have found…’ should read another study have found….
Line 99 should be … levels of spliced XBP1
Line 100-102, is same study referenced in line 100, so sentence should be rephrased to reflect this
Line 110 should read ‘a transgenic rat model…’
Line 115…should read ‘activated ATF6 translocates to the nucleus…’
Line 116 - 117…should read ‘and activate gene transcription of proteins including ER chaperones that increase ER protein folding….’
Lines 126 – 128…please rephrase sentence
Line 135 – 136…should read ‘…cytoplasmic domain of ATF6 translocates’
Line 137…’Our study…’ Not sure what study is referred to here
Lines 145 – 150…rephrase to show logic and avoid repetitions
Line 152…UPS is a kind of compound….?
Line 156…’it is involved in…’ what is?
Line 159… pls use more appropriate references
Line 179…’rhodopsin T17M mutant’ is sufficient
Line 184…axa?
Line 196 – 199…sentence too complex, please rephrase
Line 212…’misalignment of the ER membrane’ do the authors mean retrotranslocation to the cytosol?
Line 217…’mannan’ do the authors mean mannose?
Line 217… authors should include also describe briefly process of retrotranslocation of misfolded proteins to conclude this paragraph. The description of the ubiquitin enzymes should form a new paragraph. Perhaps a brief introduction to the process of ubiquitination would be appropriate to start this new paragraph, with brief mention of the specific roles of the E1, E2 and E3 enzymes, and how process primes misfolded/unfolded proteins for degradation by the 26S proteasome
Line 220 – 226…authors should Make a very clear distinction between the yeast and human E3 uniquitin ligases. The sentences could be clearer and more coherent.
Line 226 – 228…the link between the UPR effectors and the UPS needs to be better presented
Line 231…’Researchers have found’…could be replaced by ‘Studies have shown…’
Line 250…’ubiquitous life phenomenon’… authors should please rephrase
Line 262 – 264…’…and are endogenous mechanisms…’ please clarify
Line 277 – 280…sentence too complex, please check punctuations and simplify
Line 281…’researchers have…’ perhaps could be replaced with ‘Studies have shown…’
Line 296 – 298…Please rephrase sentence and include sources/references
Line 317…’we found that…’ could be replaced with ‘It was reported that/It has been shown that….’
Line 326 - 329…. Please rephrase sentence, use an active voice…. ‘The MAPK pathway activates ERK…’
Line 333…Please rephrase sentence
Line 335…’more and more…’ could be replaced with ‘increasing number…’
Line 340…. avoid repetition. Autophosphorylation of IRE1 occurs on the cytoplasmic region
Line 358…’Then…’ could be replaced by ‘Thus…’
Line 366 – 368…. Please rephrase the sentence
Line 368…the word death, could be replaced by the phrase cell death
Line 370….’one study…’ could be replaced It was shown
Line 381…please rephrase sentence
Line 386…’The latest discovery…’ could be replaced by ‘ A recent study…”
Line 399 – 404…please rephrase and simplify sentence
Line 405…’…between these 2 systems…’ please specify what systems you refer to
Line 413…’…managing the health of the cell…

Figure legends should be provided - these should succinctly explain the figures.

Reviewer 3 ·

Basic reporting

The review outlines the response of cells to stress by exploring endoplasmic reticulum stress and protein degradation systems. It deals with the complexity of the regulation of the systems involved and the role of other proteins in mediating or suppressing the activity of the protein degradation system.

There are some minor aspects that require attention including, lack of standard referencing with superscript used in a number of places, lack of space between the full-stop and the start of a new sentence, and the explanation of all acronyms when they are first used.

Line 81: should this read endoplasmic reticulum stress, autophagy and/or ubiquitin?

Use of casual language is also evident Line 54 ‘ER is broken’, line 95 ‘genes that are generally expressed’. In response to what stimulus?

Other aspects which are more significant to address include the study design and the conclusions drawn from the review.

Experimental design

The article content is within the Aims and sscope of the Journal. The authors explain the methodology used as a review of academic articles. If this has been chosen as the methodology it is more usual to put much greater detail in the text and include keywords used, number of articles identified and number of articles used for the review. This can then be regarded as a systematic review. If this is a narrative review then there is little merit in proposing a specific methodological approach as it is not possible for the reader to scrutinize the method and to be confident that the methods were comprehensive and unbiased.

Validity of the findings

The authors state in line 81 and 82 that, we propose a future research direction based on the articles. In the conclusion the authors state that. We should strive to inhibit diseases by simultaneously regulating autophagy and Ub-proteosome in order to achieve cellular homeostasis. There is clearly merit in this approach in some specific instances but the authors themselves comment on the varied role that activation of the systems have in various ophthalmic diseases (see lines 178 – 182) so this statement should be qualified with a more specific statement regarding the proposed future direction. In reality the whole system is driven by the ‘stress’ and it would be worthwhile considering how to reduce the stress rather that modify the ‘stress response’.

Additional comments

This is an interesting area and one that has worthwhile content. The authors have discussed complex cellular regulatory pathways relating to the stress response and have focussed this on the endoplamic reticulum, ubiquitin-proteosome and autophagy pathways. However, in its present form it is quite unwieldy and the authors should consider a more structured approach to the different section as there is currently a lot of overlap. It may be appropriate to consider a more substantial section on the introduction and regulation of the 3 systems and then deal with some of the ophthalmic diseases in a more deliberate way rather than weave them into the regulatory description of the endoplasmic reticulum stress pathways The complexity is covered well but would benefit from further scrutiny of the language used, the formatting of the sections to reduce the overlap and the consolidation of the ways in which the various ophthalmic conditions are identified to facilitate engagement with the text.

.

---

## Round 0.2 · accepted · Accept

Thank you for attending to the suggestions of the reviewers. I think the article has benefitted from your patient attention to these points and I am happy therefore to accept the review in this form.